# The Impact of Information Technology and Sustainable Strategies in Hotel Branding, Evidence from the Greek Environment

**Sotirios Varelas** [1] , **Panagiota Karvela** [2] **and Nikolaos Georgopoulos** [2,*]

1   Department of Tourism Studies, Economics, Business and International Studies, University of Piraeus, 185 34 Pireas, Greece; svarelas@webmail.unipi.gr
2   Department of Business Administration, Economics, Business and International Studies, University of Piraeus, 185 34 Pireas, Greece; pamina@karvela.com
*   Correspondence: ngeorgop@unipi.gr; Tel.: +30-210-414-2421

**Abstract:** Information Systems, Sustainable Development and Hotel Branding are the main study areas of this specific research. In recent years, many technologies have been which have started to be implemented in the hospitality industry, but the full potential of these technologies has not yet been analyzed in order to understand how they could help companies to develop branding strategies and increase the customer's loyalty and business performance. Moreover, due to the pandemic crisis due to COVID-19, hotel companies have to manage a new crisis, which affects both the economic and social pillars of their sustainable development. Therefore, the hospitality industry must be more flexible in adopting information technologies and sustainable philosophies, firstly for their own survival and secondly to strengthen their position inside the task environment. The methodology approach was based both on the literature review and qualitative methodology, which was conducted through questionnaires. The aim of this research is to analyze the above concepts and see how this combination could lead the hotel companies to develop a competitive advantage inside their task environment. The findings of this research concern the region of Greece. They reveal that there is a strong correlation among the above concepts, and each company should start to adopt sustainable philosophy, use information technologies, and develop new branding strategies.

**Keywords:** hotel branding; hospitality; information technology; sustainable development; competitive advantage; performance; sustainable strategy

## 1. Introduction

Tourism is perhaps the most important service industry in the world, because of the number of people it employs, and because of the consequences and impacts it has on the social and economic development of regions and countries [1]. Tourism is also one of the greatest prospects for wealth creation and jobs in all countries of the world [2]. Over the last three decades, the tourism industry has been growing rapidly, becoming one of the most key sectors in the services sector at present.

The rapid growth in the industry has led to the important growth and development of hospitality companies, which gain their income from the tourism services. Dominant among these companies are hotel companies, which have experienced a rapid development, both in terms of the number of hotel units and beds. The main goal of the hotels is, of course, to increase their development and profit creation, which is largely achieved through the delivery of quality services.

The main pillar of tourism development in Greece is mainly hotel companies. The last decade has seen a significant increase in quantity and quality in terms of hotel capacity in Greece. Typically, the number of beds has increased by 20%, while luxury 5-star hotels, as a share of the total units, have doubled by 15.2% from 8% [3]. In Greece, there are

about 40,000 units of main and supplementary tourist accommodation (10,000 main hotel accommodation) at present.

The COVID-19 pandemic and the social distancing measures that were imposed to deal with it dealt a severe blow to the tourism industry, one of the most important sectors of the Greek economy. In particular, the turnover of hotels was almost zero in April–June (compared to an average drop of 23% for the rest of the business sector), while the third quarter saw an average drop of 65% (compared to 12% for the rest of the business sector) [4].

The importance and the contribution of tourism and hotel companies in the Greek economy require the thorough study of factors that will give them a competitive advantage during these critical years of the COVID-19 pandemic. Information Systems and Information Technology can support hotel companies' strategies through innovative entrepreneurship, providing advantages in maximizing long-term profits, facing pandemic uncertainty and accelerating competition.

This study will attempt to approach the tourism product as a complex "intangible experience", as Mill and Morrison [5] (p. 457) called it, with many factors reshaping and influencing it [6], such as brand creation and the provided quality of services, supported by Information Systems (IS) and sustainable development practices. Moreover, this research will attempt to theoretically and methodologically examine the role of Information Systems (IS) and sustainable development practices in branding as a strategic role of hotel business management. Each of the above concepts are explored in this paper, specifically inside the Greek environment.

The goal of this paper is to analyze the above concepts according to the international literature review and find out how the connection between the above concepts could lead hotel companies to develop branding strategies and create a competitive advantage. The research concerns the region of Greece, and the paper reveals a strong correlation among these concepts. Moreover, to address these issues, a multi-dimensional questionnaire was developed by the authors aside from the literature review, in order to collect and analyze data on how managers and employees perceive sustainable development and the use of Information Systems inside their companies. Data were analyzed based on Resource-Based View Theory (RBV).

This paper is comprised of an Introduction, containing a brief description of the paper, followed by the Theoretical Background, where the authors try to analyze the meaning of concepts by conducting an extensive literature review. Then, in the Conceptual Framework, authors develop their way of thinking of how to deal with the concepts. In the Methodology, the authors develop their hypotheses to create the questionnaire, and after that is the Analysis, where all the collected data were analyzed. In the Results and Discussion, all the analyzed data provide the authors with useful information to discuss. To sum up, "Conclusion and Limitations" is the final chapter, that authors state the conclusions from this research, describe the limitations of this paper and declare the need for future research, in order to better examine the three pillars of sustainability.

## 2. Theoretical Background

The strategy and, more broadly, the strategic management, of the hospitality industry and hotel companies is at low levels globally. Strategic research, especially in the field of hospitality and hotel businesses, is almost non-existent compared to other fields, while the number of international academics in strategic tourism management is low. Compared to the field of strategy, the number of research studies that approach the creation of branding is even smaller, especially with the support of the Information Systems and sustainable development practices. One exception is the holistic approach [7] in the handbook of Strategic Brand Management: Building, Measuring and Managing Brand Equity. The American Marketing Association defines a "brand" as [8] (p. 123) "a name, a term, a design, a symbol, or any other attribute, which identifies with the product or service of an organization or individual and distinguishes it from the other markets".

Branding is particularly critical for services, and especially for hosting companies [9]. The well-designed application of branding can create significant added value for hotel companies, which explains the significant increase in such ventures in the field of tourism [10].

It is a well-known fact that hotel managers internationally pay special attention to the quality of services and customer satisfaction, in combination with the concept of branding, since the brand offers the customer the appropriate information about the product or service [11].

### 2.1. Hotel Branding

According to Kapferer [12], a brand as a concept is when a company differentiates its products/services from the competitors in its task environment. In the tourism and hospitality industry, it is critical to develop brand management [13]. The manager is the one who creates and communicates the concept of brand to customers [14]. Brand is defined as "the unique set of brand associations that the "brand strategist aspires to create or maintain, which imply a promise to customers from the organization" [15]. The role of a brand manager is to develop and build the long-term brand process, which develops the durability and consistency of brands, and define brand meaning as "a reflection of internal and external stakeholders' mindset about a brand" [16,17].

At present, the tourism and hospitality industry is becoming increasingly competitive; as a result, the branding process is becoming more complex [13]. Brown et al. [18] state that the creation of a strong brand is the result of marketing engagement by the customers and brand managers. Therefore, there is a co-creation process, which involves the company managers and the active engagement of clients [19]. At present, due to the COVID-19 pandemic, there is a huge boost in brand co-creation. The lockdowns which have occurred in every region and country, causing global pause in tourism and hospitality services, have created many problems for these sectors and companies, affecting their brands [13].

The above problems, which have been caused by the pandemic, have created a new travel trend, where neighborhoods have earned their position on the travelling map. Therefore, hotel brands are trying to increase customer engagement by localizing their marketing strategies [13]. Moreover, due to the pandemic, all the processes were computerized and digitized in fewer than 6 months [20]. This means that brand managers have to think about how to engage smart tourism information technologies [21] and how to provide access to customers in real time [22].

Last but not least, due to the tremendous increase of the use of social media, brand managers have the chance to interact immediately with their customers and develop an honest relationship with them, creating strong emotions and strengthening their desire to visit [23].

### 2.2. Sustainability and Branding

According to the World Commission on Environment and Development, sustainability can be defined as development which meets the present needs of society without risking the ability of future generations to satisfy their own needs [24]. Innovations which concern the environment could be used as a tool for companies to find opportunities to reduce costs and gain an eco-friendly reputation [25]. Therefore, marketing strategies play a significant role in promoting such policies, which can be supported by brand management in order to promote the value of sustainability for their customers, consumers, and other stakeholders [26].

Hotel companies consumes a high number of natural resources and energy and produce many wastes, such as food waste, with their operations affecting the sustainability of the natural environment. Final consumption, which is the final stage of food supply chain, is responsible for as much as 40% of the total food losses [27]. Recent studies show that, in developed countries, food waste occurs in the final stage of the food supply chain. Due to the current situation of our planet and the policy of the United Nations and European Union, all companies, including hotel companies, should follow the Sustainable

Development Goals (SDGs) in order to create a more sustainable environment for future generations [28].

In addition, stakeholders at present put a lot of pressure on hotel companies to be environmentally friendly and to respect regulations regarding the correct treatment of labor. In order to offer their services, companies must understand the preferences of their target groups, because each customer and traveler has completely different preferences regarding different ecological and social services and products [26].

### 2.3. Hospitality Industry and Information Technology

Information systems (IS) include a variety of information technologies, such as computers, software, databases, communication systems, the Internet, mobile devices and more, in order to perform specific tasks and inform various factors in different organizational or social contexts [29]. An Information System (IS) was created to help companies achieve their goals and objectives. Therefore, the mission of Information Systems is to improve the people's performance in an organization through the use of Information Technology [30]. Therefore, IS consists of Information Technology and humans, who are an integral part of it.

Information Technology includes:

- Data;
- Hardware;
- Software;
- Telecommunications.

Information is considered a unique and significant resource for the organization, and companies invest a large amount of money to gather, store, manage and distribute it [31]. Information is stored in large data stores, which contain a large amount of information about clients, suppliers, products, etc., and are available and useable for managers [32].

It is well-known that the Tourism Industry is dependent on information. In the field of tourism, Information Technology could play an important role in organizational process, the creation of opportunities, and equilibrium between supply and demand. The use of IT could have many benefits for the managers and companies and give them the chance to adjust to changes which occur in their own task environment [33]. Therefore, Information Systems could be hotel companies' solution to many problems inside the tourism field. Many academics and managers support the idea that the appropriate understanding of the correct use of information systems could be essential for the survivability of companies and hotels [33].

The effective and efficient use of IT in the tourism and hospitality industry is very important for tourism development [34]. IT allows for customer–management relations and supply chain management to be combined into a single source that facilitates a variety of operations—product selection, ordering, fulfillment, tracking, and payment. IT can cut costs by allowing the provider to be in direct communication with the client. Managers, in tourism companies, use IT to undertake a range of tasks that enhance the efficiency of employees in the workplace, notably online reservations [35].

There is a complex relationship between the Information Systems and the hotel industry, since IT was first used in the early 1960s [36]. Managers and employees understand the value and benefits which are derived from the use of IT [37]; however, executives often feel threatened by the complexity and rapid pace of IT evolution and the lack of reliable models to evaluate its strategic potential. Therefore, IT is necessary for the hotel strategy to increase its performance or develop a competitive advantage [38], because information technology could help hotel companies to not only improve their own operations, such as the food supply chains, but also to improve their relationships and customer service.

## 3. Conceptual Framework

At present, the concept of sustainability is becoming increasingly important and should be placed at the forefront of any company's attention. Due to the current reality

of climate change and global warming, sustainability is attracting more attention, and companies are implementing "green" policies to reduce their environmental footprint. Some companies, however, consider the cost of sustainability too high and continue to operate as they used to [39]. What they do not realize is that adopting these policies, they could have a positive impact on both companies and the other stakeholders.

In 2016, reference [39] conducted a survey comparing companies that had adopted a sustainability philosophy with others that had not. The results of his research show that many non-sustainable companies claim that following these policies is very costly, but the results also show that sustainability does not have a negative impact on the cost of goods sold, profit margin or operating costs. This should encourage companies to implement (or maintain) green initiatives, as there are many potential benefits with few drawbacks.

Sustainability helps increase a company's "internal business performance", which means that, by implementing sustainability policies in conjunction with the use of Information Systems, the company has the ability to reduce the time required to carry out the various procedures within the company. In addition, research [40] has yielded some results that support the claim that investing in sustainable practices can lead to improvements in business performance. A positive correlation was found between overall sustainable improvement practices and overall business performance. Many environmental management practices were strongly correlated with innovation performance, which can further improve economic performance [41]. During the previous years, many companies considered sustainable development as an extra and undesirable cost. This is wrong, because sustainability leads to increased business performance, which affects not only the financial but also the customer performance [42].

"Resource Based View" is a model that sees the company's resources as key to increasing the company's performance. If a resource has VRIO features, then it enables the business to have a competitive advantage [43].

According to [44], the characteristics of VRIO are:

- Value;
- Rare;
- Imitable;
- Organization.

The "RBV" model helps the company to gain a competitive advantage over its competitors.

As mentioned before, there are companies which still consider sustainability as a responsibility; studies have empirically shown that environmental performance and economic performance are positively correlated [45], and companies involved in sustainability efforts have gained legitimacy and increased market value [46].

In recent years, a focus on sustainability has been developed to help companies improve their operations, innovation, strategic development, develop a stable competitive advantage and provide sustainable values to society [47]. Recent research has used RBV analysis as a theoretical basis to support the benefits of adopting sustainable policies for business development.

RBV theory states that a company develops a competitive advantage not only because it acquires, but also because it develops, combines, and uses its natural, human and organizational resources in ways that add unique value and are difficult for competitors to emulate [48]. In addition, according to [49], the adoption of sustainable strategies could help companies to develop sustainable values and obtain a sustained competitive advantage. Moreover, the combination of sustainable and IT resources (Figure 1) allows companies to develop sustainable abilities and increase their performance.

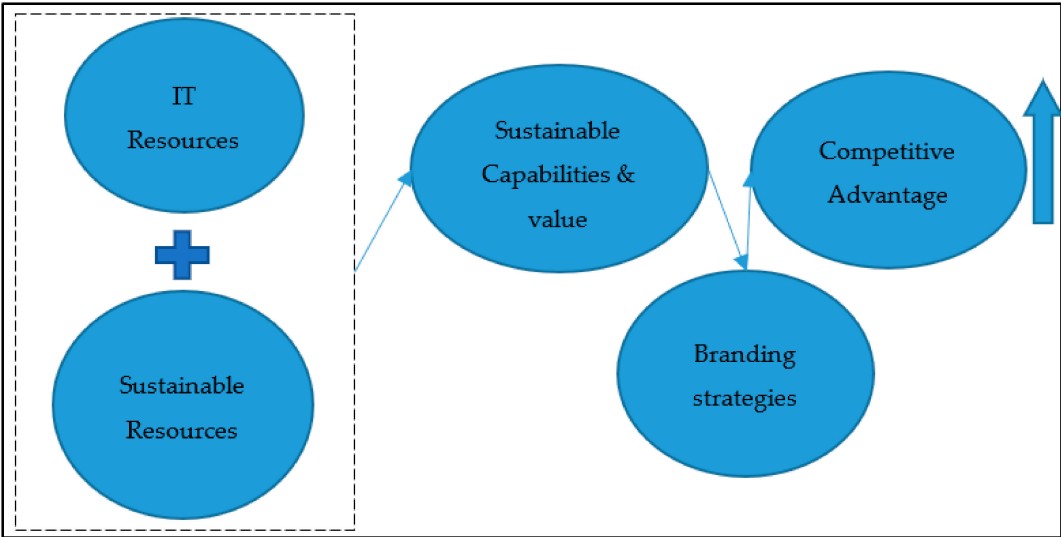

**Figure 1.** Increase in competitive advantage.

As this paper focuses on a company's competitive advantage, the developed framework shows the types of resources and abilities that can lead to its increase.

According to the above conceptual framework, hotel companies develop and use their own sustainable and Information Technology resources to make their processes and services more effective and efficient. The combination of the above resources can create sustainable abilities and, in many cases, sustainable value. Companies take advantage of these new abilities and value to develop new branding strategies. Finally, hotel managers are able to develop these strategies in order to increase their competitive advantage and strengthen their position in the market. The main concepts of this framework are further explained in the following sections.

*3.1. IT Resources*

Hospitality companies spend millions of dollars on Information Technologies each year. At present, due to digitalization, a high number of IT resources is available at market, for both back-office and front-office hotel management [50]. The most well-known are as follows [38]:

- Application Service Providers (ASP);
- Enterprise Resource Planning (ERP);
- Customer Relationship Management (CRM);
- Yield Management System (YMS);
- Property Management System (PMS).

Moreover, the most well-known and frequently used technology is that of web technology, which is followed by and combined with other technologies, such as social media. It is important to mention that, until 2015, there was only one piece of research which examined the concept of Big Data and Artificial Intelligence in the field of hospitality management [51]. This technology, as mentioned in the literature review, could soon play an important role.

*3.2. Sustainable Resources*

Due to the increasing environmental impact and lack of available resources in the global society, hotel operations need increasing support from sustainability management systems [52,53]. As the industry has a high level of resource consumption [54], operating adequate sustainability systems, such as energy renewable systems, is important for hotels to build strong business reputations. In addition, according to [55], more customers are now willing to pay more to be "green" and want to be involved in environmentally

friendly activities. In this regard, hotels operating sustainability management systems can generate a sustainability marketing plan to appeal to "green" customers. Studies have shown the increase in the competitive advantage of hotel businesses after the adoption of sustainability measures. The adoption of these "green" initiatives gives the customer a sense of responsibility for their environment and a feeling of participation in the green process.

As far as the social pillar is concerned, hotels need to be more aware about the personnel and society. Hotels and managers must pay more attention to their personnel, especially in these difficult times due to COVID-19. Moreover, personnel must be educated about sustainable policies and how to implement them, in order to more easily achieve sustainable goals. As well as the personnel, hotels must also pay attention to the other stakeholders. Finally, all stakeholders must have access information concerning social and environmental issues [56]. So, the hotel companies may increase their legitimacy, which is a very important resource for each company [57] according to the legitimacy theory.

As far as the economic pillar is concerned, hotel companies must create a large amount of money in order to adopt all these technologies; however, after their implementation, they will be able to see the economic results. Hotel companies will reduce food waste at each stage of the food chain, saving money, and, most importantly, they will increase the satisfaction of their customers, which could translate into monetary units.

### 3.3. Sustainable Capabilities and Value

The sustainable abilities that resulted from the combination of the above resources (IT and sustainable resources) in the company did not appear overnight, nor were they purchased for any monetary value, but emerged over time, such as increased knowhow, which the company acquired throughout its operation [58]. Thus, three categories of abilities were created according to the sustainable development of the company:

- Economic;
- Social;
- Environmental.

Information Systems could contribute to Sustainable Hospitality by providing better and more accurate data regarding the social pillar of sustainability. These data concern the relationship between the company and its partners and its customers. This kind of data are very difficult to quantify, so managers with this kind of information would be able to follow sustainable strategies. The application of information systems creates added value for tourism sector organizations and enhances their competitive edge.

In addition, IT allows hotel owners to improve the level of sustainable services that they offer. This could lead to an improvement in the economical pillar of sustainability, because managers are going to have a better image of the operations inside their companies and save time using the necessary IT each time. This could lead to an increase in business performance. Investment in IT may have a greater impact on the company's economic pillar than comparable spending, on either advertising or R&D, and IT investments may be more effective and efficient in improving profitability by increasing revenue through increasing customer satisfaction and customer retention ratios.

Information Technology plays a controversial role because it may have a negative impact on the natural environment at various stages in its life-cycle; however, it also presents opportunities for companies to pursue environmental sustainability by using it in different environmental procedures. "Smart hospitality" and information systems in general will provide an opportunity for hotel owners to improve the level of services that they offer to their guests, using sustainable smart and interconnected devices and applications, which belong to an information system network. The transition to green and smart hospitality is very important and can also offer several advantages to both the visitors and the owners and branding.

Therefore, when sustainable capabilities are developed from the above combination, the company is in a position to create a sustainability value for its stakeholders [49] and develop strategies which could lead to a sustained competitive advantage.

### 3.4. Branding Strategies

A brand in the hospitality sector refers to the sum of perceptions that individuals form about a service. These perceptions are based on the service's most notable characteristics, differentiating it from the other competitive products, services, and destinations [59]. According to the British Marketing Institute, the differentiation is is caused by a unique mind association that should be reproduced in line with the specific service or logotype's brand.

Brand loyalty is a critical factor that helps to create a so-called "competitive advantage" for organizations, as it develops a long-term relationship with consumers [60,61]. The branding area is one of the most effective sustainable marketing strategies [62]. Through Information Technology and Sustainability, service industries create branding, a strategy that builds long-term and value-based relations with the customers [63].

In this research, we discover that the success of a branding strategy is determined by its capacity to meet or exceed tourists' expectations through the use of information systems and sustainability [64]. Besides this, prior studies have demonstrated that hotel managers worldwide show a keen interest in service quality and customer satisfaction in combination with the organization's brand, as it is the brand that offers the most relevant information about the product/service to the visitor [65–67]. The management of branding activities is vital for improving performance, while strengthening the hotel brand means enhancing the customer's awareness and building a strong brand image in the minds of consumers.

### 3.5. Competitive Advantage

The success of IT implementation depends not only on technological resources, but also on the adoption of a management approach [38]. Thus, according to [68], the ability to organize and manage information technologies is able to generate a sustainable competitive advantage. According to the above, IT are able to improve the business performance of a hotel company, in terms of efficiency, coordination, innovation, alliance formation, etc. [69].

Therefore, the branding strategies that were developed from the above analysis could also provide hotels and managers with a unique opportunity to increase their own internal business performance and other performances, such as financial and customer performances. The increase in the above performances could provide hotels a competitive advantage inside their task environment.

## 4. Methodology

The main purpose of this research, as already explained, is whether the use of information systems based on sustainability practices by hotel companies positively affects the company brand in terms of customers. This is the main research question. The pillars on which the theory of the hotel business brand was based, and the four pillars that have been developed, were analyzed in the previous section.

Following the conceptual framework, which is analyzed in the above chapter, this research tries to find and analyze results from the Greek environment. Therefore, the methodology that was conducted was based on quantitative statistical methods research. Apart from the literature review, a questionnaire was developed by the authors in order to examine the resources, strategies and competitive advantages that hotel companies gain over time in their task environment.

The purpose of the quantitative statistical methods is to conduct an in-depth examination of the phenomena, which are affected by the environment [70]. According to [70], these phenomena may be specific strategies, green or sustainable policies, or gaining a competitive advantage or increasing business performance.

This specific methodology has some important advantages, which is the reason why it is so useful. First of all, it is able to provide valid data analysis in comparison with the quantitative method. In addition, the questionnaire provides the necessary time and freedom in the choice of data [71].

Each type of research is unique, so each questionnaire is unique. However, there are some main, basic principles that must be followed for the design and implementation of a questionnaire. In order to produce a well-designed questionnaire, researchers must dedicate thought and time to its preparation [72,73].

Moreover, the content must be related to the content and concept of the research. Therefore, it is important to identify the right target group, and to be clear about the purpose of the research [74]. Last, but not least, it is important for the target group to understand the questions and to have the knowledge needed to answer them [75].

Considering the theoretical background, two distinct hypotheses were formed for the present research.

- Ha1. The use of information systems based on sustainability practices by hotel businesses positively affects the business brand in terms of customers.
- Hb1. The commitment of hotel businesses to the use of information systems with an emphasis on sustainability positively affects the performance of the business.

The questionnaire, which was developed for the purpose of this paper, tries to combine the information technology resources with the sustainable resources and identify the branding strategies that are developed for hotel companies to develop a competitive advantage. To fir into the needs of the research, a questionnaire with 24 questions was created, with the aim of collecting the appropriate data, which were then analyzed with the help of statistical methods. The questionnaires were posted on an electronic platform and sent to Greek 4- and 5-star hotel businesses, from which more than 170 were answered. The official source of information we chose was the Hellenic Chamber of Hotels, the inventory method was chosen. The answers came exclusively from hotel business executives. Regarding the identity of the research, most of the hotel businesses that participated in the survey are very small companies, which also form the majority of hospitality companies in Greece. Specifically, 59.13% employ fewer than 10 people, 27.83% from 10 to 49 people, and the remaining 13.04% 50 or more people. The majority of participants (78.26%) employ fewer than 10 permanent employees. Regarding the room capacity of the hotels that participated in the survey, 34.78% have from 21 to 50 rooms and 30.43% up to 20 rooms, while the vast majority, 87.83%, do not belong to a hotel chain or a global hospitality brand. A total of 44.35% are classified as 3-star hotels, 20.87% are 4-star and only the 13.91% are 5-star. A total of 50.4% of the participants were owners of the hotel businesses, 24.7% were general managers and the rest were responsible for IT, sustainability, etc. It is a fact that most of Greek hotels are family businesses, so the current research is interested in their degree of adaptability of these hotels to the new digital environment. An important element of this research is that it took place during the COVID-19 crisis, meaning that all businesses that responded were in suspension, reinforcing the priority of using information systems with an emphasis on sustainability in times of crisis.

A group of questions was created regarding the use of information systems as well as hospitality information systems, with a special emphasis on sustainability. Some of the information systems noted in the questionnaires are mentioned below:

- Customer Service Systems are used in an automated electronic way;
- Customer information applications (e-concierge), enriched with activities from the wider area;
- Applications for recording and satisfying special requests (e.g., specific food, food waste, waste reduction);
- Information on, and promotion of, the portal of services related to the wider area;
- Providing Augmented Reality (AR) and Virtual Reality (VR) experiences from heritage sites in the wider area;
- Electronic Information for nearby cultural sites and attractions;

- Electronic Information on the customs and traditions of the tourist destination;
- Possibility of navigation via interactive maps of the wider tourist destination;
- Electronic Information on catering options and local events;
- Possibility to buy local products through the hotel platform;
- Information on options from Global Distribution System (GDS) activities and the creation of dynamic packages;
- Possibility of local activities offering experiences in the Central Reservation System (CRS) and the creation of dynamic packages;
- Food and Beverage Management System (F&B) connected with local producers;
- Food and Beverage Management (F&B) and Training System for sustainable supply chains;
- Menu-Suggestion Management System for the customer to reduce food waste;
- Information Systems connected to Presence Sensors for lighting and air conditioning control;
- Hotel Management Information Systems, using Master/Slave sensors;
- Building Management Systems (BMS);
- Digital or smart network of energy distribution through an information system;
- Smart Actuators and Energy Load Management through information systems;
- Information Systems for Autonomous Control, with or without natural light compensation;
- Multimedia information on the hotel for groups of people with disabilities;
- Services for the accessibility of tourist sites for groups of people with disabilities;
- Customize Customer Relationship Management (CRM) options for the selection of people from groups of people with disabilities;
- Sales and Catering Management System for groups of people with disabilities;
- Sales and Catering Management System for groups of people with special catering needs because of allergies, intolerances, etc.

Furthermore, questions about the frequency with which hotel businesses updated their information systems were asked, as well as questions about their loyalty and investment in information systems with an emphasis on sustainability. Four categories of questions were developed (Table 1), which were specific and focused on the analysis of branding and, more specifically, the following:

- Brand loyalty and knowledge;
- Brand awareness;
- Brand image;
- Consumer brand value perceptions.

**Table 1.** Hospitality sector branding axes.

| Brand loyalty & knowledge | <ul><li>Hotel positioning in the mind of the clients.</li><li>Clients' perceptions after being exposed to communication stimuli.</li><li>Clients' influence location.</li><li>Clients' opinion forming.</li></ul> | Tepeci, 1999; Maio, 2001; Hoeffler and Keller, 2003; Esch et al., 2006; Barreda et al., 2016 |
|---|---|---|
| Brand awareness | <ul><li>Easiness in defining the hotel business.</li><li>Reference to the hotel business. Awareness.</li></ul> | Tepeci, 1999; Davis, Golicic, and Marquardt, 2008; Huang and Sarigollü, 2012; Barreda et al., 2016 |
| Brand image | <ul><li>Connecting the brand to the image of the business. Connecting the image to the core competencies of the business.</li></ul> | Esch et al., 2006; Wu, Liao and Tsai, 2012; Othman and Hemdi, 2013; Mohammed and Rashid, 2018; Barreda et al., 2016 |

**Table 1.** *Cont.*

| | | |
|---|---|---|
| Consumer brand value perceptions | • Connecting the brand to clients' eligibility.<br>• Connecting the core competencies to clients' choices.<br>• Connecting the brand to synergy loyalty.<br>• Connecting the brand to sales and price rises.<br>• Connecting the brand to market share.<br>• Connecting the brand to the differentiation of the services offered.<br>• Connecting the brand to the upsurge of the customer base.<br>• Connecting the brand to the increase of the business's economic performance. | Xu and Chan, 2010;<br>Hsu, Oh and Assaf, 2012;<br>Yang and Mattila, 2016;<br>Barreda et al., 2016;<br>Raouf and Camilleri, 2019. |

In our study, we approached and analyzed hotel businesses' brands, examining the perceptions of hotel owners/managers regarding their clients–visitors. The axes we studied regarding the hospitality sector's branding emerged from detailed bibliographical research and became the objects of our questionnaire. These axes are as follows.

For the analysis of the data collected in the present research, the Factor Analysis technique was used to summarize the data, so that the relationships between the variables under consideration could be easily interpreted and understood [76]. Then, with the Confirmatory Factor Analysis, an attempt was made to confirm the specific hypotheses that were raised. Subsequently, the correlations of the main factors that arose (whose factors represented more than 90% of the variability of the respective responses on a case-by-case basis) were examined, compared each other, so that the hypotheses raised could be confirmed or rejected. Principal components' analysis was used in the current research to describe the observed variables that relate to each other and measure them. The goal of principal components analysis (PCA) is to explain most of the variability in the data, with a smaller number of variables than in the original dataset. For a large dataset with p variables, pairwise plots of each variable were compared to the other variables, but, even for moderate p, the number of these plots becomes excessive and not useful. For example, when $p = 5$, there are $p(p - 1)/2 = 20$.

Principal components analysis (PCA) finds a low-dimensional representation of a dataset, which contains as much of the variation as possible. Each of the n observations lives in a p-dimensional space, where PCA seeks a small number of dimensions that are as interesting as possible, with the concept of interesting being measured by the amount by which the observations vary along each dimension. Each of the dimensions found by PCA is a linear combination of the p features and we can take these linear combinations of the measurements and reduce the number of plots necessary for visual analysis, while retaining most of the information present in the data.

## 5. Analysis

We proceeded by creating principal components, according to the following responses:

1. Q1 (level of development of information technology communication and sustainability) from which principal component PC1 emerged (Table 2);
2. Q2 (degree of focus on digital transformation, investment in innovation, sustainable development) from which principal component PC2 emerged (Table 2);
3. Q3 (degree of use of information technologies based on sustainability, in communication with stakeholders 'tourist, TOs, travel agents, suppliers') from which principal component PC3 emerged (Table 2);
4. Q4 (Frequency of technological development and adaptation of information systems);
5. Q5 Level of use of the Hospitality Information Systems;
6. Q6 Level of use of the Hospitality Information Systems with an emphasis on Sustainability;
7. Q7.1 Brand knowledge, from which principal component PC4 emerged (Table 2);
8. Q7.2 Brand awareness, from which principal component PC5 emerged (Table 2);
9. Q7.3 Brand image, from which principal component PC6 emerged (Table 2);

10. Q7.4 Consumer brand value perceptions, from which principal component PC7 emerged (Table 1);
11. Q8 Competitive performance assessment (ROI index, revenue, bookings, repeat customers, growth rate, operation expenses, market share, online searches, positive ratings on internet platforms) from which principal component PC8 emerged (Table 2).

**Table 2.** Loadings and Proportion Variances.

|  | **PC1** | **PC2** | **PC3** | **PC4** | **PC5** | **PC6** | **PC7** | **PC8** |
|---|---|---|---|---|---|---|---|---|
| SS loadings | 1.78 | 3.00 | 3.96 | 3.49 | 5.50 | 2.78 | 8.81 | 9.06 |
| Proportion Var | 0.89 | 0.75 | 0.86 | 0.87 | 0.92 | 0.93 | 0.88 | 0.76 |

Two sources determine the number of components to select for the next stage:

- Kaiser's criterion: components with SS loadings >1. In our case, we used just one component: PC1. A more lenient alternative is Joliffe's criterion, with SS loadings > 0.7;
- Scree plot: the number of points after the point of inflexion. Assume a straight line from the first point on the right. Once this line bends considerably, count the points after the bend up to the last point on the left. The number of points is the number of components to select. On the group of scree plots below (Figure 2), we define a threshold instead of the "bend criterion", which is equal to 1 and relates the SS loadings (the horizontal without dots line). We select the number of principals, counting the points, which are greater than one (above the horizontal line). The example here is probably very simple, given that one component was finally chosen for each case. Based on both these criteria, we are able to go ahead and select the definitive number of components.

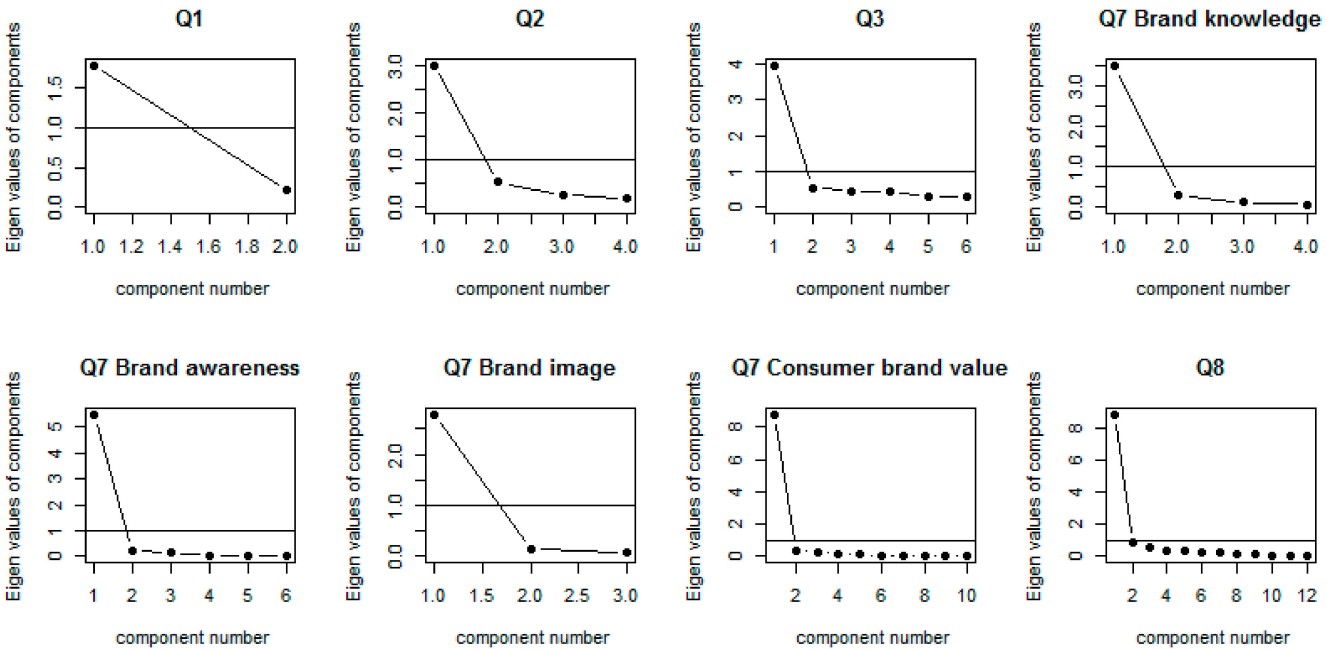

**Figure 2.** Screen plots.

Observing the Table 2, the first row contains the sum of squared loadings which are equal to the first eigenvalue. This value should be greater than one. The SS loadings are also provided on the screen plots below, as the start point counting from the right side.

The second row in Table 3 relates the total variance explained by each linear component. For component PC1, the proportional variance tells us that 89% of the variance in responses can be explained by this component, component PC2 can explain 75% of the

responses' variance, and so on. For all cases, the proportion variance is greater than 70%, giving quite a high explanatory potential for each principal component.

**Table 3.** Correlation Matrix in regard to the level of development of information technology communication and sustainability.

| Parameter1 | Parameter2 | r | CI Low | CI High | t | *p*-Value |
|:---:|:---:|:---:|:---:|:---:|:---:|:---:|
| Q1 | F7.1 | 0.644925 | 0.457988 | 0.777239 | 6.143478 | 0.0000 |
| Q1 | F7.2 | 0.570802 | 0.360024 | 0.726138 | 5.060967 | 0.0002 |
| Q1 | F7.3 | 0.5804 | 0.372453 | 0.732854 | 5.18877 | 0.0001 |
| Q1 | F7.4 | 0.562125 | 0.348852 | 0.720042 | 4.94809 | 0.0002 |

## 6. Results and Discussion

### 6.1. First Research Hypothesis

Investigating the first research hypothesis, the responses to questions 1 to 2, 5 (where Q5 refers to the number of information systems per business), as well as 6 (where Q6 also refers to the number of information systems using sustainability), were used. These were examined in relation to question 7.

More specifically, for questions 1 to 3, and question 7, the principal components derived from the respective analysis were used to examine the research hypotheses.

Thus, based on the principal component that emerged from the Q1 responses (PC1), we estimated the correlation and then tested whether this correlation was statistically significant or not, in relation to the Q7 components (PC4–PC7). The results are given on the table below (Table 3). Where r is the correlation coefficient, a low and high CI are the upper and lower limit of the confidence interval (95%), t is the observed value of the *t*-statistics, and *p*-value is the probability of obtaining results that are at least as extreme as the observed results of a statistical hypothesis test, assuming that the null hypothesis (there is no correlation) is correct.

Given that all p-values are less than 0.10 (or 10%), we can reject the null hypothesis that there is no correlation. Instead of this, we are able to say that there is a positive correlation (see r coefficient), which is statistically significant.

The same conclusions emerge when looking at Tables 4 and 5, concerning the principal components from questions 2 and 3, respectively.

**Table 4.** Correlation Matrix in regard to the degree of focus on digital transformation, investment in innovation, sustainable development.

| Parameter1 | Parameter2 | r | CI Low | CI High | t | *p*-Value |
|:---:|:---:|:---:|:---:|:---:|:---:|:---:|
| Q2 | F7.1 | 0.666859 | 0.487874 | 0.792033 | 6.514906 | 0.0000 |
| Q2 | F7.2 | 0.583541 | 0.376537 | 0.735045 | 5.231293 | 0.0001 |
| Q2 | F7.3 | 0.572964 | 0.362817 | 0.727653 | 5.089479 | 0.0002 |
| Q2 | F7.4 | 0.562115 | 0.348839 | 0.720035 | 4.947964 | 0.0002 |

**Table 5.** Correlation Matrix in regard to the degree of use of information technologies based on sustainability in communication with stakeholders.

| Parameter1 | Parameter2 | r | CI Low | CI High | t | *p*-Value |
|:---:|:---:|:---:|:---:|:---:|:---:|:---:|
| Q3 | F7.1 | 0.624721 | 0.430827 | 0.763481 | 5.824478 | 0.0000 |
| Q3 | F7.2 | 0.575334 | 0.365883 | 0.729313 | 5.120918 | 0.0001 |
| Q3 | F7.3 | 0.57736 | 0.368508 | 0.73073 | 5.147945 | 0.0001 |
| Q3 | F7.4 | 0.514553 | 0.288661 | 0.68618 | 4.368729 | 0.0013 |

Continuing with the first research hypothesis, we conclude with the examination of relations between Q7 and Q4, Q5, Q6. The respective results are presented in Tables 6–8. Again, based on the p-values and the respective coefficients of correlation, there is a positive and statistically significant correlation between all of them (significance level of 10%).

**Table 6.** Correlation Matrix with regard to the frequency of technological development, and adaptation of information systems.

| Parameter1 | Parameter2 | r | CI Low | CI High | t |
|---|---|---|---|---|---|
| Q4 | Q7.1 | −0.43765 | -0.62982 | −0.195 | −3.54355 |
| Q4 | Q7.2 | −0.3833 | −0.58873 | −0.13135 | −3.02117 |
| Q4 | Q7.3 | −0.44216 | −0.63318 | −0.20036 | −3.58882 |
| Q4 | Q7.4 | −0.36775 | −0.57678 | −0.11353 | −2.87901 |

*6.2. Second Research Hypothesis*

Finally, regarding the second research hypothesis, responses to questions Q1, Q2, Q3, Q5, and Q6 were used. These questions were examined in relation to question 8.

**Table 7.** Correlation Matrix in regard to the level of use of Hospitality Information Systems.

| Parameter1 | Parameter2 | r | CI Low | CI High | t |
|---|---|---|---|---|---|
| Q5 | Q7.1 | 0.50563 | 0.277566 | 0.679745 | 4.266636 |
| Q5 | Q7.2 | 0.440852 | 0.198807 | 0.632208 | 3.575677 |
| Q5 | Q7.3 | 0.432126 | 0.18843 | 0.625692 | 3.488439 |
| Q5 | Q7.4 | 0.328485 | 0.069221 | 0.546185 | 2.531908 |

**Table 8.** Correlation Matrix with regard to the level of use of Hospitality Information Systems with an emphasis on sustainability.

| Parameter1 | Parameter2 | r | CI Low | CI High | t |
|---|---|---|---|---|---|
| Q6 | Q7.1 | 0.605722 | 0.405604 | 0.750428 | 5.542105 |
| Q6 | Q7.2 | 0.575447 | 0.366029 | 0.729392 | 5.122416 |
| Q6 | Q7.3 | 0.61823 | 0.422175 | 0.759034 | 5.7262 |
| Q6 | Q7.4 | 0.507415 | 0.27978 | 0.681034 | 4.286903 |

It is easy to see (Table 9), based on the observed coefficient of correlation and *p*-values, that the responses of the questions Q4 and Q8 are negatively correlated, and this correlation is statistically significant (at level 10%). All the other correlations, between Q8 and Q1, Q2, Q3, Q5, Q6, are positive and significant.

**Table 9.** Correlation matrix regarding competitive performance assessment.

| Parameter1 | Parameter2 | r | CI Low | CI High | t | *p*-Value |
|---|---|---|---|---|---|---|
| Q4 | Q8 | −0.27119 | −0.50048 | −0.00635 | −2.05117 | 0.090415 |
| Q5 | Q8 | 0.401029 | 0.151892 | 0.602252 | 3.187041 | 0.024112 |
| Q6 | Q8 | 0.500359 | 0.27104 | 0.67593 | 4.207194 | 0.001904 |
| Q1 | Q8 | 0.503944 | 0.275476 | 0.678525 | 4.24755 | 0.001753 |
| Q2 | Q8 | 0.357271 | 0.101605 | 0.568668 | 2.784769 | 0.051884 |
| Q3 | Q8 | 0.533488 | 0.312407 | 0.699748 | 4.591889 | 0.000685 |

Summarizing the results of the previous tests, regarding the correlations between responses and principal components, researchers suggest that both research hypotheses are verified at a significance level of 10%. This means that: (a) the use of information systems based on sustainability practices by hotel businesses positively affects the business brand in terms of customers, (b) the commitment of hotel businesses to the use of information systems with an emphasis on sustainability positively affects the performance of the business.

According to the results, derived from the statistical analysis, the authors concluded that Information Systems are able to support the operations inside hotel companies, and could be aligned with sustainable policies to protect the environment and provide customers with the services they want. Moreover, following the branding strategies, companies

are able to increase their market share inside their task environment, and increase the performance of companies, especially the financial one, which concerns the survivability of the company. Therefore, an important contribution of this research is that managers and hotel companies, if they use all the necessary information technologies they have in their possession and apply sustainable policies in their operations, they will be able to create competitive advantage and improve their position in the task environment.

## 7. Conclusions and Limitations

According to the statistical analysis conducted in the previous chapter, researchers are able to find important clues and come to conclusions. First, information systems and sustainable development are able to influence the policies and strategies of hotel companies. The four pillars of this research, brand loyalty and knowledge, brand awareness, brand image, and consumer brand value perceptions, have a strong connection with information systems and sustainable development. Therefore, customers and clients are more willing to visit these hotels and pay for their services. Accordingly, hotel companies who have adopted these policies and strategies have a strong competitive advantage over their competitors. Therefore, the use of information systems and the adoption of sustainable practices are very important for the hotel companies because they will help them to increase their performance.

In sum, the purpose of this paper was to clarify the concepts of Information Systems, Branding and Sustainable Development, based on an international literature review. Another goal was to find the correlation between these concepts, and how this combination could lead hotel companies to develop a competitive advantage and increase their market share by developing specific branding strategies.

Therefore, there is a lack of similar research in the Greek market, which took the three pillars of sustainability and the information systems into consideration. Therefore, this specific research aimed to fill this gap, with interesting results, and to show Greek hotel companies that alignment with the new information systems and sustainable development could guide them towards a new era of innovation and customer loyalty.

Moreover, a key conclusion that emerges from the elaboration of this paper is that hotel companies in Greece have slowly begun to adopt sustainability policies, respecting the regulations of the European Union and the environment itself, and clients are more aware of the three pillars and are demanding that hospitality industry becomes more aware.

As far as the limitations are concerned, there is a significant weakness in this paper concerning the way that each company perceives the concepts and relationship between the Information Systems and Sustainable Development in its internal environment. The concepts can be different for each company, which means that each company may have a unique definition of the three pillars of sustainable development. Therefore, a common theoretical framework should be defined, so that all companies can address the same variables, because some companies may pay more attention to the environmental pillar and others to the social pillar. More specific research should be conducted in the near future to examine how companies perceive the meaning of the three pillars of sustainable development, and specifically the social pillar, because it has attracted less attention from academics, and is difficult to transform into more understandable measurable units.

**Author Contributions:** Conceptualization, S.V., P.K. and N.G.; methodology, S.V., P.K. and N.G.; software, S.V., P.K. and N.G.; validation, S.V., P.K. and N.G.; formal analysis, S.V., P.K. and N.G.; investigation, S.V., P.K. and N.G.; resources, S.V., P.K. and N.G.; data curation, S.V., P.K. and N.G.; writing—original draft preparation, S.V., P.K. and N.G.; writing—review and editing, S.V., P.K. and N.G.; visualization, S.V., P.K. and N.G.; supervision, S.V., P.K. and N.G.; project administration, S.V., P.K. and N.G.; funding acquisition, S.V., P.K. and N.G. All authors have read and agreed to the published version of the manuscript.

**Funding:** This paper is one of the deliverables of the Proposal entitled "Support for researchers with emphasis on young researchers—cycle B" (Code: EDBM103) which is part of the Operational Program "Human Resources Development, Education and Lifelong Learning" which is Co-financed by Greece and the European Union (European Social Fund) and specifically of the Operation entitled "Strengthening the Corporate Identity (Branding) of Hotel Companies through IS and Sustainable Development Practices" (MIS 5050632).

**Institutional Review Board Statement:** Not applicable.

**Informed Consent Statement:** Not applicable.

**Data Availability Statement:** Not applicable.

**Acknowledgments:** This paper is one of the deliverables of the Proposal entitled "Support for researchers with emphasis on young researchers—cycle B" (Code: EDBM103) which is part of the Operational Program "Human Resources Development, Education and Lifelong Learning" which is Co-financed by Greece and the European Union (European Social Fund) and specifically of the Operation entitled "Strengthening the Corporate Identity (Branding) of Hotel Companies through IS and Sustainable Development Practices" (MIS 5050632).

**Conflicts of Interest:** The authors declare no conflict of interest.

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
