# Peer review of "The Impact of Information Technology and Sustainable Strategies in Hotel Branding, Evidence from the Greek Environment"

_sustainability, doi:10.3390/su13158543_

Round 1
Reviewer 1 Report
In the Introduction there should be a reference on the research conducted. The reader gets into the research in page 8 which is rather far in the manuscript. Also, there should be a short paragraph at the end of the introduction with short description of the chapters that follow. Furthermore, the theoretical background is extensively long and tends to present a redundancy.
The hypotheses mentioned in page 9 are not set right, there should be just two of them since the results of the research would either confirm or not each one as mentioned in couples. I mean there is no need to express the same hypothesis in positive and negative form.
In line 18 in the abstract as well as in line 365 there is a reference in qualitative approach of the methodology employed. However, a research conducted through the use of questionnaires and by performing statistical analysis holds a quantitative approach. In general there is a vague description of the research. The authors just present that it is conducted in Greece without any further presentation.
Although there is an extended reference to the statistical methods performed (PCA) there is not a relative reference to the identity of the research, of the types of the questions (not the general issues posed, the way the questions were asked) used in order to make clear the misunderstanding of qualitative and quantitative research.
Why the numbering of the components starts with Q7? It is made believe that there are things missing… If that is what the authors wish they must provide a short explanation.
Although I believe that the authors must have done a great effort in performing the research, the manuscript fails to present it. It seems as though there is a lot of literature review which, in my opinion, does not really help the reader, and the part of the primary research is shortly elaborated. Another issue is that it seems as though there were two totally different writers, it is common knowledge that in collaborated works more than one person can contribute, however in this case it is rather obvious in not a good way.
The Conclusions part is very short and should be further expanded. The limitations and future research are not mentioned at all.
Author Response
Dear Reviewer,
We would like to thank you for the opportunity to revise our manuscript. We have just submitted our revision after dealing with all the comments and providing a full answer to the comments.
We recognize that the paper had some weaknesses at its first version and the extremely useful comments helped us to improve it significantly.
Honestly speaking and beyond the review process, with the extensive changes done, as research team we have a feeling of personal satisfaction of tranforming your work to a better paper when you read it one last time before resubmission.
We hope that the revised version will meet the expectations of yours and reviewers.
Please find bellow our changes.
In the Introduction there should be a reference on the research conducted. The reader gets into the research in page 8 which is rather far in the manuscript.
Change done from line 66-77.
Also, there should be a short paragraph at the end of the introduction with short description of the chapters that follow. Furthermore, the theoretical background is extensively long and tends to present a redundancy.
Change done from line 78-86.
The hypotheses mentioned in page 9 are not set right, there should be just two of them since the results of the research would either confirm or not each one as mentioned in couples. I mean there is no need to express the same hypothesis in positive and negative form.
Change done from line 403-406.
In general there is a vague description of the research. The authors just present that it is conducted in Greece without any further presentation.
Change done from line 417-428.
Although there is an extended reference to the statistical methods performed (PCA) there is not a relative reference to the identity of the research, of the types of the questions (not the general issues posed, the way the questions were asked) used in order to make clear the misunderstanding of qualitative and quantitative research.
Change done from line 378-387 to quantitative statistical methods.
Why the numbering of the components starts with Q7? It is made believe that there are things missing… If that is what the authors wish they must provide a short explanation.
Change done from line 510-531 with adaptation throughout the research analysis.
Although I believe that the authors must have done a great effort in performing the research, the manuscript fails to present it. It seems as though there is a lot of literature review which, in my opinion, does not really help the reader, and the part of the primary research is shortly elaborated. Another issue is that it seems as though there were two totally different writers, it is common knowledge that in collaborated works more than one person can contribute, however in this case it is rather obvious in not a good way.
It took place a holistic approach in order to improve the paper.
The Conclusions part is very short and should be further expanded.
Change done from line 646-650
The limitations and future research are not mentioned at all.
Change done from line 651-661

Reviewer 2 Report
The title, abstract and key words are suitable. The article is clear, legible, and suitable in terms of its structure. The paper is interesting and organized. The article is really well written, structured, easy to read and the references don't disturb the reading process. The length is another positive element; straight to the point, with enough evidences.
The theme is relevant, and it seems that the authors have made an important review of the literature. References are present and most of them are in English.
The methodology applied is consistent.
Results only consider a descriptive analysis, so which is the contribution of the paper? Make it clearer. I would suggest to write a properly written Discussion chapter and Research implications, Limitations as well.
Author Response
Dear Reviewer,
We would like to thank you for the opportunity to revise our manuscript. We have just submitted our revision after dealing with all the comments and providing a full answer to the comments.
We recognize that the paper had some weaknesses at its first version and the extremely useful comments helped us to improve it significantly.
Honestly speaking and beyond the review process, with the extensive changes done, as research team we have a feeling of personal satisfaction of tranforming your work to a better paper when you read it one last time before resubmission.
We hope that the revised version will meet the expectations of yours and reviewers.
Please find below our changes.
The title, abstract and key words are suitable. The article is clear, legible, and suitable in terms of its structure. The paper is interesting and organized. The article is really well written, structured, easy to read and the references don't disturb the reading process. The length is another positive element; straight to the point, with enough evidences.
The theme is relevant, and it seems that the authors have made an important review of the literature. References are present and most of them are in English.
The methodology applied is consistent.
Thank you for your kind and helpful comments.
Results only consider a descriptive analysis, so which is the contribution of the paper? Make it clearer.
Change done from line 613-622
I would suggest to write a properly written Discussion chapter and Research implications, Limitations as well.
Change done from line 623-661 and a new chapter Conclusions and Limitations added.

Round 2
Reviewer 1 Report
Τhe authors did a lot of work in reforming the manuscript. The issues set through the original comments in the first review round were answered.
Reviewer 2 Report
The authors took my comments well into account. They have responded to the main issues.